# TMBIM4 Deficiency Facilitates NLRP3 Inflammasome Activation-Induced Pyroptosis of Trophoblasts: A Potential Pathogenesis of Preeclampsia

**DOI:** 10.3390/biology12020208

**Published:** 2023-01-29

**Authors:** Yuanyao Chen, Lin Xiao, Guoqiang Sun, Min Li, Hailan Yang, Zhangyin Ming, Kai Zhao, Xuejun Shang, Huiping Zhang, Chunyan Liu

**Affiliations:** 1Institute of Reproductive Health, Tongji Medical College, Huazhong University of Science and Technology, Wuhan 430030, China; 2Department of Obstetrics and Gynecology, Maternal and Child Hospital of Hubei Province, Tongji Medical College, Huazhong University of Science and Technology, Wuhan 430030, China; 3The First Hospital of Shanxi Medical University, Taiyuan 030001, China; 4Department of Pharmacology, School of Basic Medicine, Tongji Medical College, Huazhong University of Science and Technology, Wuhan 430030, China; 5Department of Andrology, Jinling Hospital, School of Medicine, Nanjing University, Nanjing 210002, China

**Keywords:** TMBIM4, pyroptosis, NLRP3 inflammasome, preeclampsia, trophoblasts

## Abstract

**Simple Summary:**

Impaired invasion of extravillous trophoblasts (EVTs) results in inadequate remodelling of arteries and poor placentation, leading to preeclampsia (PE). Transmembrane BAX Inhibitor-1 Motif-containing 4 (TMBIM4) was found to promote the migration and invasion of human osteosarcoma U2-OS and breast cancer MCF7 cell lines. However, the effect of TMBIM4 on trophoblast biological behaviour and its relevance to PE pathophysiology remain unclear. In this study, we confirmed that TMBIM4 was highly expressed in cytotrophoblasts, syncytiotrophoblasts, and EVTs of the human placenta during early pregnancy. Moreover, TMBIM4 was found to be significantly decreased in PE placenta. Thereafter, lipopolysaccharide (LPS) treatment reduced the expression of TMBIM4 and induced NLR family pyrin domain-containing 3 (NLRP3) inflammasome activity in a first-trimester human trophoblast cell line, the HTR-8/SVneo cell line. Knockout (KO) of TMBIM4 in the HTR-8/SVneo cell line impaired cell viability, migration, and invasion, which was more severe in the LPS/ATP-treated TMBIM4-KO cell line. TMBIM4 deficiency enhanced NLRP3 inflammasome activity and promoted subsequent pyroptosis. Inhibiting the NLRP3 inflammasome with MCC950 in HTR-8/SVneo cells alleviated LPS/ATP-induced pyroptosis and damaged cell function in the TMBIM4-KO cell line. Overall, this study discovered a new PE-associated protein, TMBIM4, and its biological significance in trophoblast pyroptosis mediated by the NLRP3 inflammasome.

**Abstract:**

Impaired invasion of EVTs results in inadequate remodelling of arteries and poor placentation, leading to PE. TMBIM4 was found to promote the migration and invasion of human osteosarcoma U2-OS and breast cancer MCF7 cell lines. However, the effect of TMBIM4 on trophoblast biological behaviour and its relevance to PE pathophysiology remain unclear. In this study, we confirmed that TMBIM4 was highly expressed in cytotrophoblasts, syncytiotrophoblasts, and EVTs of the human placenta during early pregnancy. By comparing the expression levels of TMBIM4 in the placenta of women with normal-term pregnancy and PE, TMBIM4 was found to be significantly decreased in PE. Thereafter, we determined the expression of TMBIM4 in the LPS-treated first-trimester human trophoblast cell line HTR-8/SVneo (mimicking a PE-like cell model), and determined the effect of TMBIM4 on trophoblast function and its underlying mechanism. LPS treatment reduced the expression of TMBIM4 and induced NLRP3 inflammasome activity in HTR-8/SVneo cells. KO of TMBIM4 in the HTR-8/SVneo cell line impaired cell viability, migration, and invasion, which was more severe in the LPS/ATP-treated TMBIM4-KO cell line. Moreover, TMBIM4 deficiency enhanced NLRP3 inflammasome activity and promoted subsequent pyroptosis, with or without LPS/ATP treatment. The negative relationship between TMBIM4 expression and NLRP3 inflammatory activity was verified in PE placentas. Inhibiting the NLRP3 inflammasome with MCC950 in HTR-8/SVneo cells alleviated LPS/ATP-induced pyroptosis and damaged cell function in the TMBIM4-KO cell line. Overall, this study revealed a new PE-associated protein, TMBIM4, and its biological significance in trophoblast pyroptosis mediated by the NLRP3 inflammasome. TMBIM4 may serve as a potential target for the treatment of placental inflammation-associated PE.

## 1. Introduction

Preeclampsia (PE) is known as one of the most serious pregnancy-associated diseases, affecting approximately 3% to 5% of pregnant women [1]. The main clinical indicators of PE are new-onset hypertension occurring after 20 weeks of gestation along with proteinuria or multiorgan failure, such as in the liver and kidneys [2]. PE poses severe health risks for pregnant women and foetuses as it threatens their life and long-term health [3,4]. Although the pathological mechanism of PE remains elusive, poor placentation and altered local or systemic inflammation have been proposed as the main etiological factors [5].

Proper functioning of trophoblasts is critical for placental development. Extravillous trophoblasts (EVTs) migrate from trophoblast cell columns and directly invade spiral arteries through the decidual stroma, partially replacing endothelial cells. This process remodels the uterine spiral artery to increase maternal blood flow to the placenta [6]. Impaired invasion of EVTs into spiral arteries induces deficient remodelling of the arteries and inadequate placentation, which might be the fundamental cause of PE [7,8].

Impairment of trophoblast migration and invasion is closely linked to trophoblast cell death [9]. Pyroptosis is an inflammatory programmed cell death characterised by the formation of gasdermin D (GSDMD) at the plasma membrane and release of numerous intercellular components [10]. Elevated pyroptosis of the trophoblasts of PE placentas was previously identified [11,12]. Inhibition of trophoblast pyroptosis alleviates PE pathological changes in mice [13]. The NLR family pyrin domain-containing 3 (NLRP3) inflammasome, the most fully characterized inflammasome, is a cytosolic complex composed of three subunits, NLRP3, apoptosis-associated speck-like protein (ASC), and Pro-caspase-1 [14,15], that have been identified in human primary trophoblasts [16]. In response to danger signals, such as pathogen-associated molecular patterns (PAMPs) and damage-associated molecular patterns (DAMPs), the NLRP3 inflammasome is activated in human trophoblasts, which mediates the cleavage of pro-caspase-1 and contributes to GSDMD-mediated pyroptosis and the subsequent release of mature interleukin (IL)-1β and IL-18 [11,16,17]. The administration of procoagulant extracellular vesicles (EVs) was found to trigger a PE-like phenotype in C57BL/6 mice by activating the NLRP3 inflammasome in the trophoblasts of the mouse placenta, and genetic inhibition of inflammasome activation by knockout of NLRP3 or caspase-1 abolished the PE-like phenotype [18]. Therefore, understanding the underlying mechanism of excessive NLRP3 inflammasome activation in trophoblasts will aid the development of strategies for the prevention of PE.

Transmembrane BAX Inhibitor-1 Motif-containing 4 (TMBIM4), also known as Golgi anti-apoptotic protein (GAAP), is critical for maintaining cell viability and providing protection against apoptotic stimuli by modulating intracellular Ca^2+^ levels and fluxes [19,20]. Previously, the overexpression of TMBIM4 was found to promote cell adhesion and migration due to the activation of store-operated Ca^2+^ entry [21]. Recently, Almeida et al. [22] found that the overexpression of hGAAP stimulated 3-dimensional proteolytic cell invasion by enhancing mitochondrial metabolism in an intracellular hydrogen peroxide-dependent manner. However, whether TMBIM4 regulates trophoblast function and is involved in PE pathogenesis remain unclear.

In this study, the expression of TMBIM4 in PE placentas and its potential effect on trophoblast function were determined for the first time. Furthermore, the molecular mechanism by which TMBIM4 regulates trophoblast invasion and migration was determined. TMBIM4 was found to be specifically expressed in human placental cytotrophoblasts (CTBs), syncytiotrophoblasts (STBs), and EVTs. Compared with women with normal-term pregnancies, TMBIM4 was downregulated in the placenta of women with PE. By constructing TMBIM4 knockout HTR8/SVneo cell lines, TMBIM4 was found to regulate cell viability, migration, and invasion by inhibiting NLRP3 inflammasome-mediated pyroptosis.

## 2. Materials and Methods

### 2.1. Gene Expression Omnibus (GEO) Data Acquisition

TMBIM4 mRNA expression levels in the placenta of normal pregnant (NP) women and PE patients were analysed from two GEO datasets, GSE147953 and GSE75010. The expression data of GSE147953 were classified into NP (*n* = 4) and PE placentas (*n* = 4), while those of GSE75010 were classified into NP (*n* = 53) and PE placentas (*n* = 67). 

### 2.2. Sample Collection 

The study subjects were recruited at Hubei Women’s and Children’s Hospital and The First Hospital of Shanxi Medical University from 8 July 2020 to 20 January 2022. PE patients were diagnosed on the basis of the definition proposed by the International Society for the Study of Hypertension in Pregnancy (ISSHP), which includes a kind of new-onset hypertension (systolic blood pressure > 140 mmHg, diastolic blood pressure > 90 mmHg) concomitant with one or more other features, including proteinuria, disturbances of other maternal organs (including liver, kidneys, and nervous system), blood involvement, and/or placental disturbances such as restricted foetal development and/or placental blood flow ultrasound Doppler abnormalities [2]. Women with a singleton pregnancy with the condition of normal blood pressure and the absence of obstetrics or complications were placed in the normal pregnancy (NP) group. All placental tissues (1 × 1 × 1 cm^3^) were harvested from the centre of the maternal placental surface. Then, the samples were washed with phosphate-buffered saline (PBS), and split in half: one segment was kept at −80 °C and the other segment was fastened in paraformaldehyde before embedding. After matching for age, gestational age at delivery, and BMI, three samples from the NP and PE groups were chosen for the subsequent experiments. The clinical characteristics of the six participants are summarized in Appendix A. Early placental tissue was derived from the placenta of a normal pregnant woman who voluntarily terminated her pregnancy for non-medical reasons at 6 to 10 weeks of gestation. 

### 2.3. Cell Culture and Treatments

HTR-8/SVneo cells were maintained in a complete medium that incorporated RPMI-1640 medium (RPMI-1640; Gibco, Carlsbad, CA, USA), 10% foetal bovine serum (Gibco, USA), and 1% penicillin-streptomycin (BioFrox, Guangzhou, China) in a 5% CO_2_ incubator at 37 °C. HTR-8/SVneo cells were treated with 100, 200, and 400 ng/mL lipopolysaccharide (LPS, Sigma-L2880) for 24 and 48 h, and then added to 5 mM ATP for 45 min. The PBS of the same capacity was used as the vehicle control. The HTR-8/SVneo cells were divided into the following groups: WT group (wild-type HTR-8/SVneo cells); KO group (stable TMBIM4-knockout HTR-8/SVneo cells); WT+LPS/ATP group (wild-type HTR-8/SVneo cells treated with LPS/ATP); KO+LPS/ATP group (stable TMBIM4-knockout HTR-8/SVneo cells treated with LPS/ATP); and KO+LPS/ATP+MCC950 group (stable TMBIM4-knockout HTR-8/SVneo cells treated with LPS/ATP and 0.01 μM MCC950). ATP and MCC950 were purchased from MedChemExpress (Monmouth Junction, NJ, USA). 

### 2.4. Vectors and Transfection 

In this study, human *TMBIM4* small guide RNAs (sgRNAs) were designed by managing the online design tool CRISPR (http://crispr.mit.edu, accessed on 18 January 2019) by inputting the target exon sequence. The CRISPR/Cas9 vector pX330 plasmid was digested with BbsI (New England Biolabs, Ipswich, MA, USA), and three PX330-TMBIM4 plasmids encoding different sgRNAs targeting TMBIM4 were constructed and verified via sequencing analysis. The three TMBIM4-specific gRNA sequences were as follows: 5′-ATCGAGGAGCGAGGGTACCG-3′, 5′-GTTGAAGTCGTCCTCGATCG-3′, and 5′-CATTCGGATGTGCACGGTGG-3′. Thereafter, the PX330-TMBIM4 plasmids were transfected into HTR-8/SVneo cells using Lip3000 transfection reagent (Invitrogen, Carlsbad, CA, USA). Successful transfection of the PX330-TMBIM4 plasmids was visually confirmed using a fluorescence microscope. After monoclonal amplification, culture and sequencing were carried out, and identification was performed using PCR and genomic DNA sequencing.

### 2.5. Immunohistochemistry 

After 4% paraformaldehyde fixation, the placental tissues were dehydrated and paraffin-embedded, and 5 μm thick sections were prepared. After tissue deparaffinization and hydration, peroxide intimal activation was performed with 3% hydrogen peroxide. Then, they were incubated overnight at 4 °C with anti-TMBIM4 antibody (1:200, ER43683, HUABIO, Hangzhou, China). The control group included slides incubated with rabbit IgG in a 1% BSA/PBS (negative control). The sections were then incubated using biotinylated secondary antibodies and stained for detection using DAB (G1022, Servicebio, Wuhan, China). Images were taken with a microscope (Ming Mei Shot, Guangzhou, China). Positive signals were analysed using ImageJ software.

### 2.6. Immunofluorescence 

Paraffin-embedded tissues were sectioned at 5 μm thickness. The slide was subsequently dewaxed, hydrated in graded alcohol, and sealed with 5% goat serum for 45 min. Primary antibodies against human CK7 (1:200, Service, GB12225) and human TMBIM4 (1:200, ER43683, HUABIO, Hangzhou, China) were incubated at 4 °C overnight. Afterwards, the sections were coated with 1/500 secondary antibody (Servicebio, Wuhan, China). 4′,6′-diamidino-2-phenylindole (DAPI) was used to stain nuclei. Photographs were taken with a microscope (Apotome3, Zeiss, Oberkochen, Germany).

### 2.7. Immunoblotting

Lysates of placental tissues and cells were prepared in RIPA lysis buffer, which contained protease inhibitors (Beyotime, Shanghai, China). After the placental tissue and cells were homogenised, the resulting product lysates were centrifuged for 20 min at 12,000 rpm. Proteins (20 μL, 60 μg of total protein) were added on 10% SDS-PAGE gels (P0012AC, Beyotime, Shanghai, China) and placed onto polyvinylidene difluoride (PVDF) membranes. Then, the membranes were blocked with 5% non-fat milk for 60 min at room temperature, and the membrane was incubated with primary antibodies, such as anti-TMBIM4 (1:1000, AP53325, ABCEPTA, Suzhou, China), anti-NLRP3 (1:1000, AG-20B0014-C100, AdipoGen, Listar, Switzerland), anti-IL-1β (1:1000, YT5201, Newark, ImmunoWay), anti-IL-18 (1:1000, 60070-1-1g, Proteintech, Wuhan, China), anti-caspase-1 (1:1000, 22915-1-AP, Proteintech, Wuhan, China), anti-GSDMD (1:1000, 20770-1-AP, Proteintech, Wuhan, China), and anti-GAPDH (1:1000, 60004-1-Ig, Proteintech, Wuhan, China), at 4 °C overnight. After washing with TBST (20 mM Tris–HCl, 150 mM NaCl, and 0.1% Tween-20) three times, the membranes were then incubated with corresponding horseradish peroxidase (HRP)-goat anti-mouse secondary antibodies (1:5000, BL001A, Biosharp, Guangzhou, China) or HRP-goat anti-rabbit secondary antibodies (1:5000, BL003A, iosharp, Guangzhou, China) for 2 h at 25 °C. After using an enhanced chemiluminescence detection kit (SQ201, Omni-ECL, Shanghai, China), the membranes were examined by a Bio-Rad imaging system (Bio-Rad, Hercules, CA, USA) to detect antibody signals. Densitometric analysis of the blots was performed using ImageJ software. GAPDH was used as the internal standard.

### 2.8. Cell Viability Assay

Cell viability, which could be detected by using a cell counting kit-8 (CCK8, 521942, Biosharp, Guangzhou, China), was an indicator of cellular state. In this experiment, cells were seeded in a 96-well plate at a density of 4× 10^3^ cells/well and cultured for 24 and 48 h. Five repeating groups were measured for each group and time point. Ten microlitres of CCK8 solution was mixed into each well, and the plate was incubated in a cell incubator for 1 h. Optical density (OD) at 450 nm was measured using an enzyme labeller (BioTek Instruments Inc., Winooski, VE, USA).

### 2.9. Transwell Assay 

Cell migration and invasion abilities were measured using the transwell method. For the migration assay, 200 μL of the WT cell and TMBIM4 KO cell suspensions (1 × 10^5^/mL) without FBS were added to each well of the apical transwell chamber (Corning, New York, NY, USA), while 800 μL of RPMI 1640 medium with 15% FBS was added to each well of the basolateral chambers. The appropriate chemical solution (LPS/ATP, MCC950, PBS) for each group was added to the apical transwell chamber and mixed. After incubation for 24 h, the cells were fixed for 30 min at room temperature and then washed with PBS. Non-moving cells in the apical transwell chamber were gently wiped away with a cotton swab. Then, the transwell chamber was stained with 0.1% crystal violet for 30 min and washed 3 times with PBS. Images were captured using a DP Manager-70 microscope (Olympus, Tokyo, Japan). Five fields per sample were randomly selected to calculate the penetrating cells using ImageJ. As regards the invasion assay, 70 µL of diluted Matrigel (1:8, 354234, BD Biocoat, Horsham, PA, USA) was added to the middle of the apical chamber and then placed in the cell incubator for about 2 h until rehydration of the Matrigel. All subsequent operations were consistent with those of the migration assay described above. 

### 2.10. Terminal Deoxyribonucleotidyl Transferase-Mediated Deoxyuridine Triphosphate-Digoxigenin Nick End Labeling (TUNEL) Assay

After 48 h of chemical treatment (LPS/ATP, MCC950, PBS), TUNEL staining was used to detect the death of WT and TMBIM4-KO HTR-8/SVneo cells. All operations were performed on the grounds of the manufacturer’s instructions (MK1020/MK1025, BOSTER, Pleasanton, CA, USA). Images were taken with a microscope (Apotome3, Zeiss, Oberkochen, Germany). ImageJ was used to count TUNEL-expressing cells.

### 2.11. Lactate Dehydrogenase (LDH) Assay 

In the assay, WT and TMBIM4 KO HTR-8/SVneo cells were cultured in a medium containing the corresponding chemical solution (LPS/ATP, MCC950, PBS) for 48 h. The supernatant was collected and assessed to measure LDH activity on the basis of the manufacturer’s instructions (A020-2-2, Jiancheng, Nanjing, China). The optical density values of the different groups of LDH were compared with a microplate reader (BioTek Instruments Inc., Winooski, VE, USA). The LDH concentration was calculated according to the formula.

### 2.12. Enzyme-Linked Immunosorbent Assay (ELISA)

The assay-requested culture supernatants were collected from WT and TMBIM4-KO HTR-8/SVneo cells which handled with the corresponding chemical treatment (LPS/ATP, MCC950, PBS) for 48 h, and to determine the expression levels of IL-1β (P273883, R&D, Minneapolis, MN, USA) and IL-18 (P635270, R&D, Minneapolis, MN, USA) in culture supernatants using ELISA kits according to the manufacturer’s instructions.

### 2.13. Statistical Analysis

All data processing was performed using GraphPad Prism 8 statistical software. All measurement information was presented as mean ± standard deviation (SD). Two groups were compared using Student’s t-test. Differences between multiple groups were compared using one-way analysis of variance (ANOVA), followed by post hoc Tukey ‘s multiple comparisons. Binary logistic regression (OR) was used to evaluate the association of placental TMBIM4 level with the risk of PE, taking maternal age and pre-pregnancy BMI (BMI) as possible confounding factors for PE [23,24]. A *p*-value < 0.05 indicates that the difference is significant.

## 3. Results

### 3.1. Expression of TMBIM4 in the Villi and Decidua during Early Human Pregnancy 

To determine whether TMBIM4 is expressed in trophoblasts, TMBIM4 expression and position in the villus and decidua tissue in the first trimester were analysed using immunohistochemistry and immunofluorescence. The floating villi consist of two layers of trophoblast cells, inner and outer, in which the cells in the outer layer fuse with each other and the intercellular boundary disappears, called the STBs, and the inner layer of cells is clearly defined and is called the CTBs. During development, CTBs divide to increase the number of cells and create STBs. After the third month of pregnancy, CTBs mainly disappear and only STBs remain [25]. As shown in Figure 1A, immunohistochemical analysis revealed significant TMBIM4 expression in the villi during early pregnancy. TMBIM4 was mainly expressed in cell column trophoblasts, CTBs, and STBs. The villi attached to the decidua produce proliferating cell columns that lead to differentiated EVTs. EVTs gradually replace the vascular epithelium, thereby remodelling spiral arteries into larger conduits that deliver low-pressure, high-volume blood flow to the growing foetus [5]. TMBIM4-positive signals were mainly distributed in the endothelial layer of the spiral artery in the first-trimester decidua tissue (Figure 1B). Furthermore, immunofluorescence staining revealed the co-localisation of TMBIM4 with the trophoblast marker CK7 in the villi and decidua (Figure 1C), verifying the primary expression of TMBIM4 in CTBs, STBs, and EVTs.

### 3.2. Expression of TMBIM4 Is Significantly Decreased in PE Placentas

The enriched expression of TMBIM4 in trophoblasts suggests that TMBIM4 may play a role in the occurrence of trophoblast dysfunction-associated pregnancy complications. To determine the potential relationship between TMBIM4 and the pregnancy-associated disease PE, the mRNA levels of TMBIM4 in the placenta of four preeclamptic patients and four normal pregnancy (NP) women from the GEO dataset GSE147953 and 67 preeclamptic patients and 53 normal pregnant women from the GEO dataset GSE75010 were evaluated. The TMBIM4 expression level was found to display a declining trend in PE placentas compared to the NP placentas from the GSE147953 dataset (Figure 2A). Further, TMBIM4 was significantly reduced in the placentas from the PE group compared to the NP group from the GSE75010 dataset (Figure 2B). The GSE75010 dataset is the largest and most complete dataset to date, with gene expression data and clinical data for NP and PE placental samples. TMBIM4 expression was reduced in the PE group of the GSE75010 dataset after adjusting for maternal age and BMI (Figure 2C). The protein level and distribution of TMBIM4 in the collected NP and early-onset PE samples were assessed. Immunohistochemical analysis of the placental sections revealed predominant expression of TMBIM4 in the STBs in normal placentas at term and a markedly weaker signal of TMBIM4 in PE placentas (Figure 2D–H). Immunoblotting further revealed a reduction in TMBIM4 protein levels in the three PE placentas compared with the three NP placentas (Figure 2I,J). In conclusion, TMBIM4 is significantly downregulated in PE placentas, highlighting its potential role in PE pathogenesis.

### 3.3. TMBIM4 Knockout Decreases the Cell Viability, Migration, and Invasion of the HTR-8/SVneo Cell Line with or without LPS Treatment

Low-dose LPS has long been used to induce PE [26]. To evaluate the expression of TMBIM4 in response to LPS stimulation in trophoblasts, a first-trimester human extravillous trophoblast cell line, HTR-8/SVneo, was treated with different concentrations of LPS for 48 h followed by 5 mM ATP for 45 min to mimic the inflammatory environment in preeclamptic trophoblasts. Immunoblot analysis revealed a significant decrease in TMBIM4 protein expression in HTR-8/SVneo cells exposed to 200 and 400 ng/mL LPS (Figure 3A,B). NLRP3 inflammasome activation is a major downstream inflammatory response in trophoblasts and is implicated in the pathogenesis of PE [27,28]. In addition to the decrease in TMBIM4, immunoblot analysis revealed that the protein levels of NLRP3 and the product of NLRP3 inflammasome activation, caspase-1 p20, increased in an LPS dose-dependent manner (Figure 3C,D). In the subsequent study, 400 ng/mL LPS was used to mimic the inflammatory environment of preeclamptic trophoblasts. 

To determine the potential function of TMBIM4 in trophoblasts, an HTR-8/SVneo cell line with TMBIM4 knockout (KO) was constructed using CRISPR/Cas9 to mimic the reduction in TMBIM4 in PE trophoblasts (Figure 3E). The KO efficacy of TMBIM4 at the translational level (Figure 3F–G), which could be used for subsequent studies, was confirmed through immunoblotting. The viability of TMBIM4 KO stable HTR-8/SVneo cells significantly decreased after 24 h of culture and further decreased after 48 h (Figure 3H). Transwell assays revealed that TMBIM4 knockout significantly reduced the migration and invasion ability of HTR-8/SVneo cells (Figure 3I–K). After LPS/ATP treatment, the impairment of cell viability, migration, and invasion was more severe in TMBIM4 knockout HTR-8/SVneo cells than in cells subjected to other conditions (Figure 3I–K). Thus, TMBIM4 knockout can inhibit the viability, migration, and invasion of HTR-8/SVneo cells with or without LPS treatment. 

### 3.4. TMBIM4 Knockout Augments the Formation of NLRP3 Inflammasome and Pyroptosis of the HTR-8/SVneo Cell Line

Owing to the negative relationship between TMBIM4 expression and NLRP3 inflammasome activation in LPS-induced inflammatory response in HTR8/Svneo cells, we focused on the role of TMBIM4 in the NLRP3 inflammasome to determine the underlying mechanism of TMBIM4 in the regulation of trophoblast function. We determined the effect of TMBIM4 knockout on the protein levels of NLRP3 and downstream molecules, with or without LPS/ATP treatment. TMBIM4 knockout significantly upregulated the expression of caspase-1 p20, cleaved IL-1β, cleaved IL18, and GSDMD proteins in cell lysates. Further, TMBIM4 knockout significantly upregulated the expression of NLRP3 upon LPS/ATP treatment, and downstream molecules caspase-1 p20, cleaved IL-1β, cleaved IL18, and GSDMD became more pronounced upon LPS/ATP treatment (Figure 4A–F). As excessive NLRP3 inflammasome activation contributes to pyroptosis, we assessed the effect of TMBIM4 knockout on trophoblast pyroptosis. Based on the TUNEL assay, the number of TUNEL-positive cells significantly increased in the TMBIM4 KO group compared to the WT group. Notably, this change was more obvious with LPS/ATP exposure (Figure 4G,H). Detection of the release of lactate dehydrogenase (LDH), a marker of plasma membrane pore formation, enables the assessment of the extent of pyroptosis [29]. Based on our results, TMBIM4 KO cells significantly increased the content of LDH compared to WT cells with or without LPS/ATP exposure (Figure 4I), and the levels of the pro-inflammatory cytokines IL-1β and IL-18 were significantly increased in the cell media supernatant of the LPS/ATP-treated TMBIM4 KO group compared with the WT, TMBIM4 KO, and LPS/ATP-treated WT groups (Figure 4J,K). These observations suggest that TMBIM4 knockout reinforced LPS-induced NLRP3 inflammasome activity and pyroptosis of HTR-8/SVneo cells.

To further validate whether the regulatory effect of TMBIM4 on NLRP3 inflammasome activity is involved in the development of PE, we performed immunoblotting to compare the protein levels of TMBIM4, NLRP3, and IL-1β in the placentas of NP and early-onset PE groups (the same samples as used in Figure 2I). As shown in Figure 4L–O, the decreased expression of TMBIM4 was accompanied by a marked elevation in NLRP3 and IL-1β protein levels in the placenta of the PE group compared to that of the NP group. Thus, the excessive NLRP3 inflammatory responses mediated by TMBIM4 insufficiency and the subsequent pyroptosis of trophoblasts may be a potential contributor to PE.

### 3.5. Inhibition of NLRP3 Inflammasome Relieves the Inflammatory Cascade and Pyroptosis of the TMBIM4-KO HTR8/SVneo Cell Line 

We proceeded to determine whether impaired trophoblast function caused by TMBIM4 deficiency was dependent on NLRP3 inflammasome activation and pyroptosis. The NLRP3 inflammasome-specific inhibitor MCC950 was used to explore whether inhibition of NLRP3 inflammasome activity is sufficient to counteract the effects induced by TMBIM4 knockout. After MCC950 treatment, the attenuated cell viability, migration, and invasion ability induced by the knockout of TMBIM4 in LPS/ATP-treated trophoblasts were recovered (Figure 5A–C,F). By evaluating the cell death rate, LDH, and the concentrations of IL-18 and IL-1β in the cell supernatant, we found that TMBIM4 deficiency enhanced the pyroptosis of LPS/ATP-treated HTR8/SVneo cells, which was clearly alleviated by MCC950 (Figure 5D,E,G–I). In summary, the mitigation of pyroptosis and impaired trophoblast function based on the inhibition of the NLRP3 inflammasome in TMBIM4-KO HTR8/SVneo cells indicated that TMBIM4 may maintain trophoblast function by regulating NLRP3 inflammasome activity. 

## 4. Discussion 

Excessive placental inflammation due to infections and non-infectious stimuli at the maternal–foetal interface accounts for various adverse pregnancy outcomes, including PE [30,31]. The inflammatory response in trophoblasts, the main cell type of the placenta, is not only a crucial strategy to induce trophoblastic death that affects placental development, but it is also possible to cause systemic damage in the mother through secreted inflammation [11,28]. However, little is known about the potential mechanisms underlying trophoblast inflammation. In this study, we discovered a new PE-related protein, TMBIM4, that was significantly downregulated in the placenta of women with PE. TMBIM4 was mainly expressed in the trophoblasts of the human placenta, and TMBIM4 deficiency in the trophoblast cell line markedly enhanced NLRP3 inflammasome activity and promoted subsequent pyroptosis, ultimately disrupting trophoblast viability, migration, and invasion, and might contribute to the pathological process of PE (Figure 6).

In human osteosarcoma U2-OS and cervical cancer HeLa cell lines, TMBIM4 is considered a novel regulator of focal adhesion dynamics, cell adhesion, and migration by activating store-operated Ca^2+^ entry, thereby stimulating calpain 2 activity to increase the cleavage of focal adhesion proteins [21]. Almeida et al. [22] confirmed that TMBIM4 overexpression induces cell invasion and matrix degradation in the U2-OS and human breast cancer MCF7 cell lines. In addition, U2-OS cells overexpressing TMBIM4 displayed stronger adhesion and colonisation abilities in vivo. TMBIM4-induced cell invasion was proposed to promote mitochondrial respiration and increase intracellular ROS. It is well known that insufficient EVT migration/invasion is related to obstetric syndromes, including PE [7]. In this study, TMBIM4 was specifically enriched in CTBs, STBs, trophoblast cell columns, and EVTs in the first-trimester human placenta. The prominent expression of TMBIM4 in trophoblasts indicates its potential role in the regulation of trophoblast function. By constructing a TMBIM4-KO HTR8/SVneo trophoblast cell line, TMBIM4 deficiency was found to impair cell viability, migration, and invasion. Thus, the role of TMBIM4 in regulating trophoblast migration and invasion was validated.

By analysing the RNA-seq data from two GEO datasets, *Tmbim4* RNA levels were found to be significantly decreased in the placenta of women with PE compared to those of women with normal pregnancies. We further verified that the mRNA and protein levels of *Tmbim4* were downregulated in the placenta of PE patients after delivery in comparison with those in normotensive pregnant women. These data indicate that the reduced level of TMBIM4 expression in the placenta was associated with PE. As an inflammation inducer, LPS is recognised by TLR4 receptors on the surface of trophoblasts and stimulates intracellular inflammatory responses, such as the NF-κB signalling pathway and NLRP3 inflammasome activation [28,32,33,34]. In this study, 200 ng/mL and 400 ng/mL LPS significantly induced the activation of the NLRP3 inflammasome but reduced TMBIM4 expression in HTR8/SVneo trophoblasts, indicating that the downregulation of TMBIM4 in the placenta of patients with PE may be related to the induction of cytotoxic factors present at the maternal–foetal interface. 

Current studies suggest that the development of PE is closely associated with microbial infections. Part of these microbes are capable of shedding LPS, which can stimulate an innate immune response [35,36]. Mechanistically, LPS can activate TLR4, leading to the production of pro-inflammatory cytokines, which are involved in the pathogenesis of PE [37,38]. In addition, elevated ATP level was considered as one of the main causal factors in the pathogenesis of PE [39,40,41]. It is worth noting that LPS/ATP is a classical activation combination for NLRP3 inflammasomes [29,32,33,42]. Thus, we used combined exposure of LPS/ATP to trophoblasts to explore the potential role of TMBIM4 on regulation of NLRP3 inflammasome activity in PE.

The negative relationship between TMBIM4 and NLRP3 inflammasome activity in LPS/ATP-treated trophoblasts indicates that TMBIM4 might be involved in LPS/ATP-induced activation of the NLRP3 inflammasome. Excessive NLRP3 activation in the trophoblasts of mouse placenta was proposed to trigger a PE-like phenotype in mice by impairing trophoblast differentiation [18,43]. Elevated IL-1β and IL-18 production by NLRP3 inflammasome-mediated pyroptosis of human primary trophoblasts has been implicated in the induction of placental inflammation and PE syndrome [11,44].

In the current study, the protein levels of the NLRP3 inflammasome components, NLRP3, and the products of the NLRP3 inflammasome, including caspase-1, cleaved IL-1β, and cleaved IL-18, were upregulated in the TMBIM4-KO HTR8/SVneo cell line. The product of the NLRP3 inflammasome caspase-1 cleaves GSDMD to induce cell pyroptosis and, thus, secrete intracellular contents, including IL-18 and IL-1β [10,15]. Accordingly, the indicators of pyroptosis, including the cell death rate, LDH activity, and the content of IL18 and IL-1β in the cell supernatant, were higher in TMBIM4-KO cells than in WT cells. Under LPS/ATP induction, NLRP3 inflammasome activity and pyroptosis were further aggravated in the TMBIM4-KO trophoblasts. Treatment with the NLRP3 inhibitor MCC950 relieved the LPS/ATP-induced pyroptosis of TMBIM4-KO trophoblasts. Thus, TMBIM4 deficiency leads to trophoblast pyroptosis via activation of the NLRP3 inflammasome. As a highly inflammatory form of programmed cell death, pyroptosis not only results in trophoblast cell death, compromising the impaired ability of trophoblast invasion and differentiation, but also releases intracellular contents, including DAMPs, into the maternal–foetal interface to induce local and systemic inflammation. Taken together, we speculate that the decreased expression of TMBIM4 in trophoblasts may be responsible for the occurrence of PE by exacerbating trophoblast pyroptosis under infection or non-infection stimuli. 

## 5. Conclusions

In summary, we identified a new regulator of the NLRP3 inflammasome in trophoblasts. When trophoblasts are exposed to pregnancy-incompatible factors, such as LPS, the expression of TMBIM4 is suppressed, thereby aggravating NLRP3 inflammasome-mediated inflammation, resulting in trophoblast dysfunction and maternal systematic syndrome. Accordingly, TMBIM4 may serve as a potential target for the treatment of placental inflammation-associated PE by preventing hyperactivation of the NLRP3 inflammasomes in the placenta. However, further investigations are needed to confirm this hypothesis.

## Figures and Tables

**Figure 1 biology-12-00208-f001:**
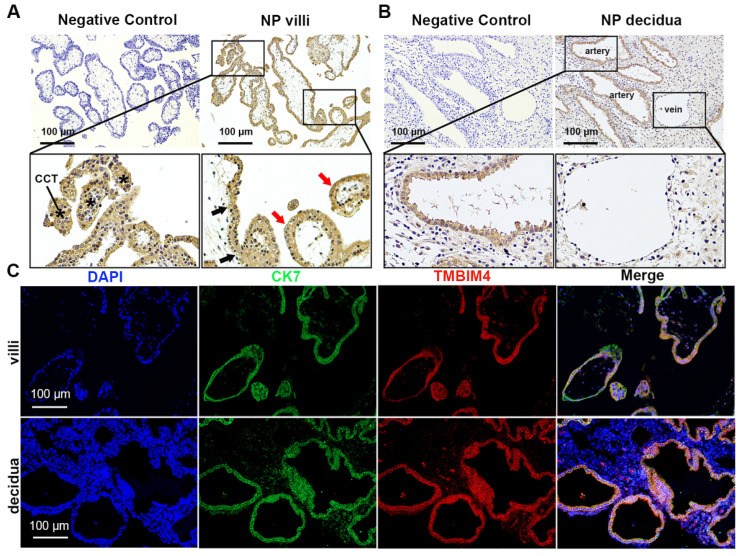
TMBIM4 expression in the trophoblasts of early pregnancy villi and decidua tissue. (**A**,**B**) Expression of TMBIM4 in early placental villi and decidua. Black arrow: CTB (cytotrophoblasts); red arrow: STB (syncytiotrophoblasts); *: CCT (cell column trophoblast); NP: normal pregnancy. (**C**) Localization of TMBIM4 in the first-trimester villi and decidua was detected via immunofluorescence (red); CK7 was used as the trophoblast marker (green), while DAPI was used to detect the nucleus (blue). Bar = 100 μm.

**Figure 2 biology-12-00208-f002:**
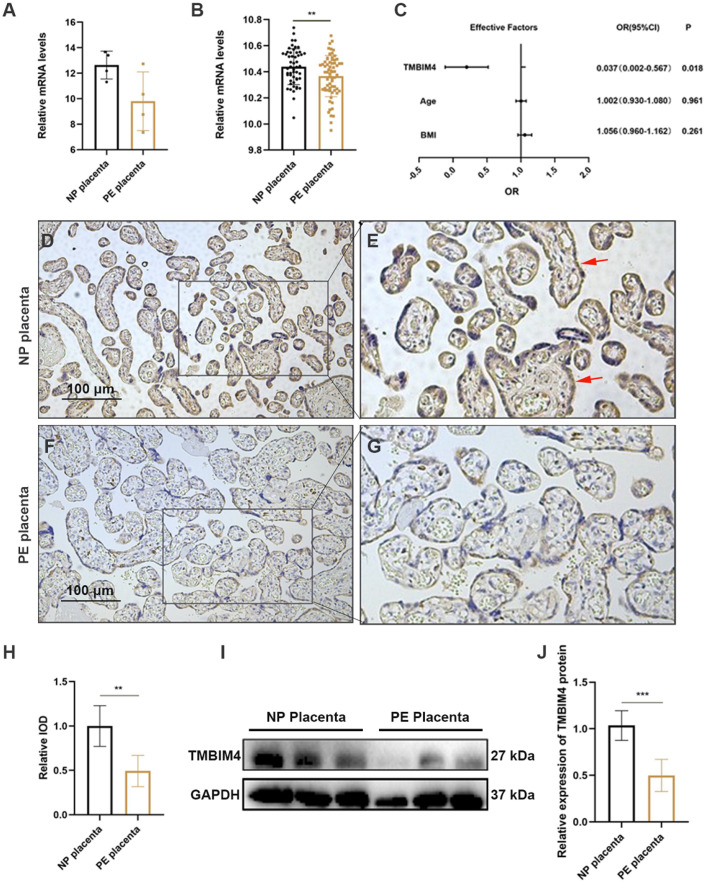
Decreased TMBIM4 levels in PE placentas. (**A**) Column plots showing normalized placental TMBIM4 mRNA expression from PE patients (*n* = 4) and NP women (*n* = 4) based on the analysis of the GEO dataset GSE147953. (**B**) Column plots showing the expression of TMBIM4 mRNA in PE (*n* = 63) and NP (*n* = 53) placentas. (**C**) Adjusted ORs with 95% CIs for the relationship between PE and TMBIM4 levels, which was adjusted for maternal age and BMI. (**D**–**H**) Representative immunohistochemical image showing TMBIM4 protein expression and localization in placentas from the NP and PE group (*n* = 3 in each group, scale bars: 100 μm). Red arrows refer to the positive signals on STBs. (**I**,**J**) The expression of placental TMBIM4 protein in NP and PE; graphs showing the results of quantitative densitometry analysis of TMBIM4. The results are presented as mean ± SD. ** *p* < 0.01; *** *p* < 0.001; Bar = 100 μm.

**Figure 3 biology-12-00208-f003:**
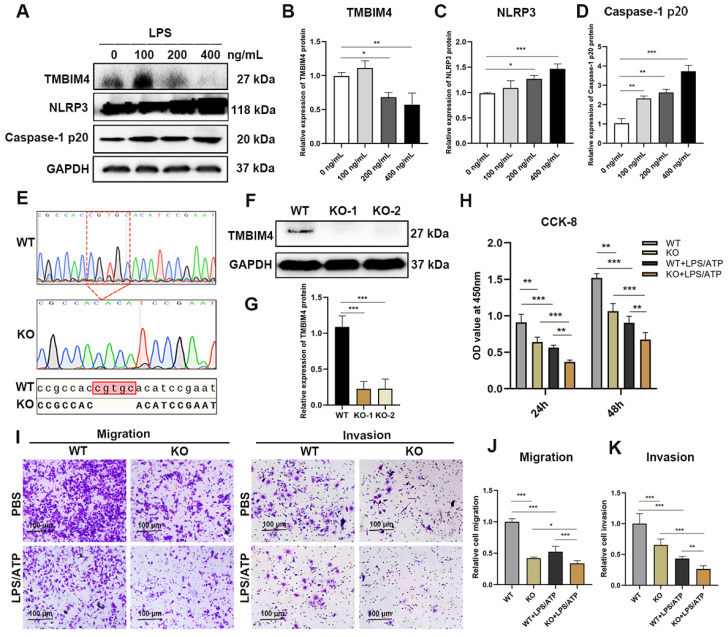
TMBIM4 knockout decreased the cell viability, migration, and invasion of HTR-8/SVneo cells. (**A**) HTR8/SVneo cells were treated with the indicated concentration of LPS for 48 h followed by 5 mM ATP for 45 min. Cell lysates were immunoblotted with an anti-TMBIM4 antibody, anti-NLRP3 antibody, anti-caspase-1 antibody, and anti-GAPDH antibody. (**B**–**D**) Graphs showing the results of quantitative densitometry analysis of TMBIM4 (**B**), NLRP3 (**C**), and caspase-1 p20 (**D**). (**E**) Representative results of DNA sequencing from TMBIM4 knockout (KO) HTR8/SVneo cells with frameshift mutations compared with wide type (WT) cells. (**F**,**G**) Knockout efficiency of TMBIM4 KO HTR8/SVneo cells based on immunoblot. (**H**) Viability of TMBIM4 KO cells compared to WT cells without or with LPS/ATP exposure based on CCK-8. Each assay was independently conducted at least 5 times. (**I**–**K**) Migration and invasive ability of WT and TMBIM4 KO stable HTR8/SVneo cells without or with LPS/ATP exposure based on the transwell assay and statistical analysis. The results are presented as mean ± SD. * *p* < 0.05; ** *p* < 0.01; ** **p* < 0.001; Bar = 100 μm.

**Figure 4 biology-12-00208-f004:**
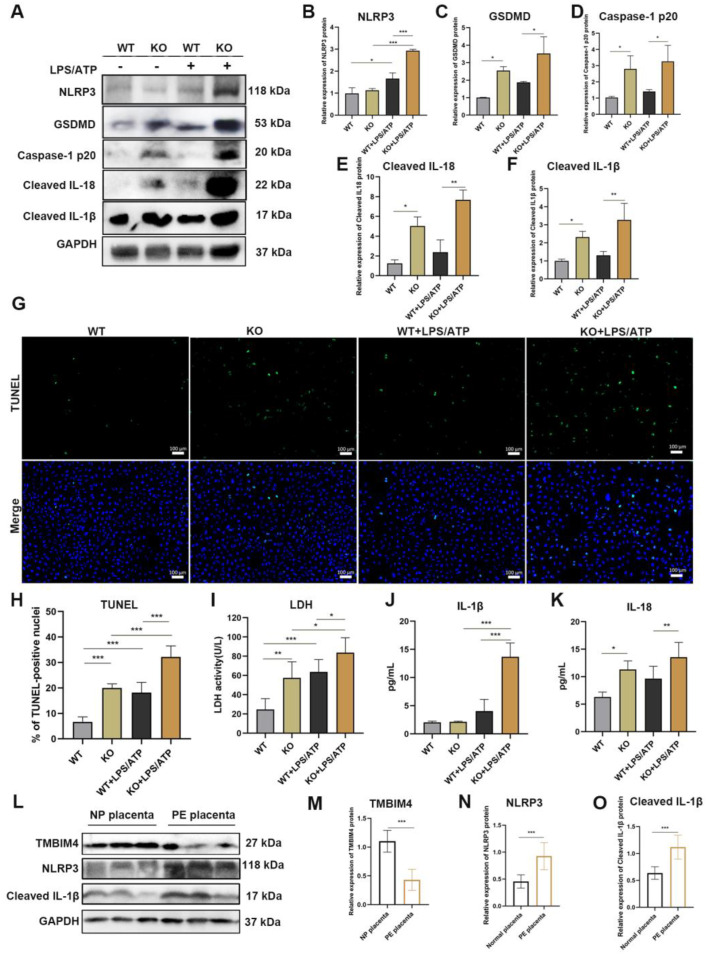
TMBIM4 deficiency enhances LPS-induced NLRP3 inflammasome activation and pyroptosis. (**A**) Immunoblot images of NLRP3, GSDMD, caspase-1 p20, cleaved IL-1β, and cleaved IL-18. (**B**–**F**) Graphs showing the results of quantitative densitometry analysis of NLRP3 (**B**), GSDMD (**C**), caspase-1 p20 (**D**), cleaved IL-1β (**E**), and cleaved IL-18 (**F**) levels in cell lysates. (**G**,**H**) Representative TUNEL assay images of WT, TMBIM4 KO, LPS/ATP-treated WT, and LPS/ATP-treated TMBIM4 KO HTR8/SVneo cells (**G**); the percentage of dead cells is presented in H. Dead cells are indicated by a green signal and the cell nuclei are indicated by a blue signal (DAPI stain). (**I**) Graphs showing the content of LDH leakage into the cell culture medium in four groups. Each assay was independently conducted at least 5 times. (**J**,**K**) Production of IL-18 and IL-1β in the culture medium of the indicated cells based on ELISA. Each assay was independently conducted at least 5 times. (**L**) Immunoblot images of TMBIM4, NLRP3, and IL-1β protein expression in placental samples from the NP and early-onset PE groups (**L**). (**M**–**O**) Quantitative analysis of the TMBIM4, NLRP3, and IL-1β immunoblot bands relative to the control based on densitometry. All quantitative data are presented as mean ± SD. * *p* < 0.05, ** *p* < 0.01, *** *p* < 0.001. Bar = 100 μm.

**Figure 5 biology-12-00208-f005:**
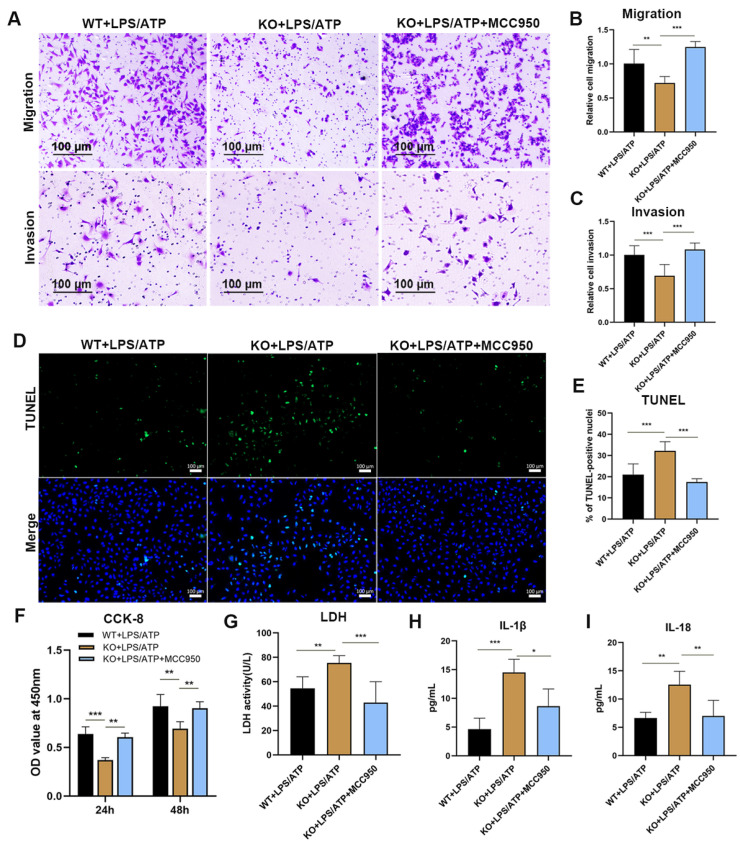
Inhibition of the NLRP3 inflammasome alleviated the inflammatory cascade and pyroptosis of TMBIM4 KO HTR8/SVneo cells. (**A**–**C**) Representative images of transwell migration and invasion in WT+LPS/ATP, KO+LPS/ATP, and KO+LPS/ATP+MCC950 cells (**A**) and qualification analysis (**B**,**C**). (**D**,**E**) Representative TUNEL assay images of WT+LPS/ATP, KO+LPS/ATP, and KO+LPS/ATP+MCC950 HTR8/SVneo cells (**D**); the percentage of dead cells is presented in E. Dead cells are indicated by a green signal and the cell nuclei are indicated by a blue signal (DAPI stain). (**F**) Viability of cells in the WT+LPS/ATP, KO+LPS/ATP, and KO+LPS/ATP+MCC950 groups based on the CCK-8 assay. Each assay was independently conducted at least 5 times. (**G**) Graphs showing the content of LDH leakage into the cell culture medium in the WT+LPS/ATP, KO+LPS/ATP, and KO+LPS/ATP+MCC950 groups. Each assay was independently conducted at least 5 times. (**H**,**I**) Production of IL-18 and IL-1β in the culture medium of indicated cells based on ELISA. Each assay was independently conducted at least 5 times. The results are expressed as mean ± SD. * *p* <0.05; ** *p* <0.01; *** *p* <0.001, Bar = 100 μm.

**Figure 6 biology-12-00208-f006:**
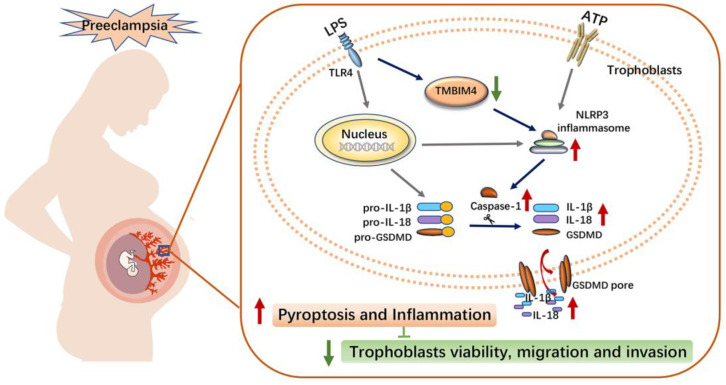
Schematic diagram illustrating the potential mechanism of action of TMBIM4 on PE pathogenesis. LPS induced the downregulation of TMBIM4 in the trophoblasts. TMBIM4 deficiency in the trophoblasts markedly enhanced the NLRP3 inflammasome activity and promoted subsequent pyroptosis, thereby disrupting trophoblast viability, migration, and invasion, and might be involved in the pathogenesis of PE.

## Data Availability

All data included in this study are available upon request by contacting the corresponding author.

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
