# Peer review of "TMBIM4 Deficiency Facilitates NLRP3 Inflammasome Activation-Induced Pyroptosis of Trophoblasts: A Potential Pathogenesis of Preeclampsia"

_biology, 2023, doi:10.3390/biology12020208_

Round 1

Reviewer 1 Report

The authors tried to show that the role of TMBIM4 on the NLRP3 inflammasomes and programed cell death pyroptosis using trophoblast cell-based assay model and the placenta of preeclamptic patients. Most of the data in this manuscript are convincing and presented by well-designed experiments. Addressing these concerns would strengthen the conclusions of the manuscript.

Major points:
1. The authors used ATP to mimic PE placenta in the manuscript. Is ATP elevated in the placenta of PE patients? Is there such evidence? Please add a logical explanation why ATP should be added. For example, in Figure 3, treatment with LPS only can decrease migration and invasion of trophoblast cells?

2. The authors used LPS to mimic PE placenta in the manuscript. The reviewer understood that treatment with LPS can stimulate NLRP3 inflammasome system and mimic PE pathology in animal models. However, is LPS involved in placental inflammation in PE patients? Please explain the justification for using LPS.

3. Please discuss how TMBIM4 regulates the inflammasome and whether it may directly inhibit it.

Minor points:
1. Line 146: please describe the company and number about LPS used in this manuscript.

2. Line 153: check the spell about the dose of MCC950.

3. Line 381: delete “NLRP3” because NLRP3 did not increase in KO cells in Figure 4B.

4. Figure 4L and M: What is the difference from protein expression in Figure 2? Are the samples different? Please add an explanation if so. In addition, molecular weight of TMBIM4 is 37 kDa?

Author Response

Thank you very much for your questions and suggestions, we have made a detailed response, the response content has been attached to the document.

Reviewer 2 Report

The manuscript deals with a very interesting topic. The publication is written correctly, but I have a few concerns.

1. There are some spelling and stylistic errors in the manuscript. The authors should correct them.

2. The bibliography includes a table with a description of the patients' BMI. It should be moved to another publication section.

3. Not all abbreviations used in the text are explained. Authors should carefully expand each of the abbreviations used in the text.

Author Response

(The authors gave the same response as above.)

Reviewer 3 Report

This submitted manuscript focused on the potential role of TMBIM4 on regulating NLRP3 inflammasome activation in vitro and in vivo.  The authors revealed that TMBIM4 was highly expressed in cytotrophoblasts, syncytiotrophoblasts, and EVTs of the human placenta during early pregnancy. Moreover, TMBIM4 was found to be significantly decreased in PE placenta.  LPS reduced the expression of TMBIM4 and upregulated the activity of NLPR3 inflammasome in vitro. Knockout (KO) of TMBIM4 impaired cell viability, migration, and invasion.  TMBIM4 deficiency enhanced NLRP3 inflammasome activity and promoted subsequent pyroptosis. Inhibiting the NLRP3 inflammasome alleviated LPS/ATP-induced pyroptosis and damaged cell function. Overall, this study discovered a new PE-associated protein, TMBIM4, and its biological significance in trophoblast pyroptosis mediated by the NLRP3 inflammasome.  In general, this manuscript was in good written, well-structured, and easy to follow.  The results were very clear and the immunofluorescent pictures were in good quality. 

One concern is that the correlation between TMBIM4 and NLRP3 inflammasome activity was not very well supported by the current findings.  KO approach is good but need to be verified by overexpression approach.  The authors should perform experiments whether TNBIM4 overexpression could mitigate the LPS-mediated upregulation on NLRP3 inflammasome activity. 

Author Response

(The authors gave the same response as above.)

Round 2

Reviewer 3 Report

The current version is fine.